# Hot Melt Extrusion-Triggered Amorphization as a Continuous Process for Inducing Extended Supersaturable Drug Immediate-Release from saSMSDs Systems

**DOI:** 10.3390/pharmaceutics14040765

**Published:** 2022-03-31

**Authors:** Huan Yu, Yanfei Zhang, Yinghui Ma, Huifeng Zhang, Chengyi Hao, Yong Zhang, Zhengqiang Li, Xianrong Qi, Nianqiu Shi

**Affiliations:** 1School of Pharmacy, Jilin Medical University, Jilin 132013, China; yuhuanjlu@163.com (H.Y.); zyf20181812@163.com (Y.Z.); mayinghui04329@163.com (Y.M.); Hfjlmu@163.com (H.Z.); yuleilei120@163.com (C.H.); 2College of Life Science, Jilin University, 2699 Qianjin Street, Changchun 130012, China; zhypharm@jlu.edu.cn (Y.Z.); lzq@jmu.edu.cn (Z.L.); 3Department of Pharmaceutics, School of Pharmaceutical Science, Peking University, Beijing 100191, China; qixr@bjmu.edu.cn

**Keywords:** hot melt extrusion (HME)-triggered amorphization, continuous manufacturing process, supersaturating amorphous self-micellizing solid dispersions (saSMSDs), extended supersaturable immediate release, “spring-parachute” processes, crystallization inhibition and stabilization of amorphism

## Abstract

Hot melt extrusion (HME), a continuous manufacturing process for generating supersaturating amorphous self-micellizing solid dispersion systems (saSMSDs), holds promise for achieving amorphization of many pharmaceutical formulations. For saSMSDs generation, HME-triggered continuous processes offer advantages over traditional non-continuous processes such as fusion/quench cooling (FQC) and co-precipitation (CP). Here we employed HME, FQC, and CP to generate saSMSDs containing the water-insoluble BCS II drug nitrendipine (NIT) and self-micellizing polymer Soluplus^®^. Scanning electron microscopy, powder X-ray diffraction, and differential scanning calorimetry results revealed that saSMSDs formed when NIT–Soluplus^®^ mixtures were subjected to the abovementioned amorphization methods. All saSMSDs outperformed crystalline NIT preparations and physical mixtures in achieving extended supersaturable immediate release states with superior solubility, “spring-parachute” process characteristics, and dissolution behaviors. Notably, Fourier transform-infrared spectroscopic results obtained for saSMSDs detected hydrogen bonding interactions between the drug and the carrier. Ultimately, our results revealed the advantages of HME-triggered amorphization as a continuous process for significantly improving drug dissolution, increasing solubility, and maintaining supersaturation as compared to traditional amorphization-based techniques.

## 1. Introduction

Immediate release, the opposite of long-term controlled release, refers not only to the rapid drug release, but also to the extended release of higher concentrations of drugs over a relatively longer period of time (1~2 days) [1,2]. Relative to conventional dosage forms and long-acting controlled-release formulations (e.g., several months’ release) [3], short-term immediate release formulations provide several potential advantages, such as improved therapeutic effect through enhanced release of insoluble drugs, improved delivery of rapid-onset therapeutics in clinical settings, and increased drug effectiveness [4,5]. Thus, it is extremely important to develop short-term immediate-release formulations of insoluble drugs by investigating initial drug release after dosing, evolving the release during the dissolution period (for optimal biological absorption), and terminal release as the final step in the process. Nevertheless, although the latter two release phases are essential, they are often ignored in studies of drug release profiles and mechanisms for use in guiding the development of innovative immediate-release formations [1,6].

Based on high-throughput evaluations, more than 50% of active pharmaceutical ingredients (APIs) exhibit water solubility issues, resulting in their assignment to the biopharmaceutical classification system II (BCS II) group of compounds, which possess natural properties of high permeability and low water solubility. Supersaturating drug delivery is a promising strategy for enhancing the bioavailability of drugs delivered orally as components of supersaturating drug delivery systems (SDDS) [7]. Consequently, SDDS for short-term drug applications has been utilized to enhance the dissolution/bioavailability of immediate-release formulations of BCS II drugs [8,9,10]. In fact, many formulations of BCS II (or BCS IV) drugs can generate supersaturated systems upon exposure to the gastrointestinal lumen [11]. This concept requires that supersaturation must be established and maintained for an adequately long period of time during the release period in order to support drug absorption during biologically relevant timeframes [12,13]. With regard to SDDSs, the “spring-parachute” theory is used to describe change profiles associated with supersaturated states. For supersaturable controlled-released formulations, the generation of supersaturation (“spring”), a non-equilibrium state, can be achieved for amorphous materials [14], nanoparticles [15], co-crystals [16] and crystalline salts [17]. Subsequently, the maintenance of a supersaturated amorphous state (“parachute”) can often be achieved by creating amorphous solid dispersions (ASD) through the introduction of polymers [18], surfactants [19], and cyclodextrins [20], such as hydroxypropyl methylcellulose, poloxamer, polyvinyl alcohol, hydroxypropyl cellulose, polyvinylpyrrolidone, d-α-tocopheryl polyethylene glycol 1000 succinate, and Soluplus^®^ [21,22,23]. Importantly, maintaining an amorphous state enhances API release by reducing the lattice energies of crystalline compounds to achieve a disordered molecular arrangement [24]. However, the amorphous state is an inherently metastable state, such that APIs tend to return to stable crystalline forms by undergoing phase transitions. Such phase transitions, which can occur during API release in liquid environments as well as in solid environments during storage, can offset any advantages of amorphization and supersaturation [25]. To address these challenges, newer types of ASDs have been developed recently that hold promise for controlling or extending short-term immediate drug release [26].

For the purpose of immediate drug release, in this work, supersaturating amorphous self-micellizing solid dispersion systems (saSMSDs) were developed based on the incorporation of self-micellizing polymers [18,27]. Unlike other conventional polymers, self-micellizing polymers can act as carrier matrix polymers that reinforce API release due to their amphipathic properties that support the formation of nano-micelle structures in aqueous solution. One such self-micellizing polymer, polyvinyl caprolactam-polyvinyl acetate–polyethylene glycol graft copolymer (Soluplus^®^) (Figure 1A), has been incorporated into numerous drug release systems based on ASDs [28], nanoparticles [29], nanomicelles [30], 3D printed scaffolds [31], and self-microemulsifying systems [32]. As compared with non-self-micellizing amorphous solid dispersions, saSMSDs generated using self-micellizing polymers, including poly [MPC-co-BMA] [27], Kollophor^®^ RH40 [33] or Soluplus^®^ [29], provide improved dissolution and solubilization of BCS II drugs.

Many methods have been adopted to achieve amorphization of immediate release ASDs, with such methods generally based on non-continuous and continuous processes. Non-continuous processes, which involve co-precipitation, freeze-drying, spray-drying, and melt-quench cooling [34], can be time-consuming and often utilize environmentally unfriendly organic solvents. By contrast, hot melt extrusion (HME), a typical continuous manufacturing process, offers several advantages over non-continuous processes [35]. For example, organic solvent-free HME minimizes the occurrence of API hydrolysis while also providing a green manufacturing option. HME, a continuous pharmaceutical process, has gained traction in various pharmaceutical fields associated with the manufacturing of implants, films, tablets, and so on [36,37,38] as a process that offers advantages over non-continuous processing techniques. For example, HME can be used to prepare carrier/drug materials at temperatures above the glass transition temperature (T_g_) to achieve effective molecular-level mixing of drugs and polymers. Such molecular mixing may transform a drug from a crystalline state to an amorphous state that may alter immediate release behaviors (e.g., dissolution or solubility) [39,40]. During continuous extrusion process, rheological profiles such as viscosity as a function of shear rate, material properties before and after the extrusion/injection, and tests for analyzing functional injection/extrusion, may affect the product properties of saSMSDs independently from the kind of materials used [41]. In addition, HME allows the preparation of comparatively thermodynamically stable products that can be easily scaled up by the pharmaceutical industry [42]. Nonetheless, the impacts of manufacturing methods derived from continuous and non-continuous process on supersaturating release characteristics of saSMSDs and amorphous APIs are unclear, especially when employing relatively new polymer carrier matrix formulations (e.g., self-micellizing Soluplus^®^). The insufficiency is unfavorable for guiding the future industrialization of saSMSDs manufacturing techniques.

Nitrendipine (NIT) is a calcium channel blocker that promptly lowers blood pressure in patients with mild to moderate hypertension while sustaining this effect long-term after administration [43]. However, NIT is a poorly water-soluble API (~2 μg/mL) that is viewed as a model BCS II drug. The aim of this work was to compare short-term release profiles of the water-insoluble drug NIT incorporated within saSMSDs prepared using a continuous manufacturing process (HME) versus non-continuous manufacturing processes such as fusion/quench cooling (FQC) and co-precipitation (CP), followed by assessments of the saSMSD/NIT preparation physicochemical properties using scanning electron microscopy (SEM), powder X-ray diffraction (PXRD) and differential scanning calorimetry (DSC). In order to assess the short-term immediate release of NIT from these saSMSDs throughout the entire drug release process, all three drug release stages were investigated. First, initial drug release was evaluated in several media using dissolution measurements [44]. Next, evolved drug release was monitored based on semi-continuous monitoring in order to demonstrate the benefits of continuous preparation processes for elevating the so-called “spring-parachute” process. Thereafter, terminal release was evaluated by measuring 24-h and 48-h release concentrations to compare the continuous and non-continuous saSMSD/NIT preparation processes on the maintenance of APIs in amorphous states.

In order to better understand observed NIT release behaviors, several underlying molecular mechanisms were explored using various experimental assessments. First, crystallization inhibition experiments were conducted to assess the inhibitory effects of self-micellizing polymer on the crystallization of supersaturated amorphous drugs during the evolved release period, since drug release and crystallization during this period are competitive. Next, phase solubility and Gibbs free energy were evaluated to identify whether self-micellizing polymer solubilization effects led to improved drug release. Finally, molecular interactions were studied using Fourier transform-infrared (FT-IR) spectroscopy to understand superior release kinetics induced by the continuous preparation process (as compared to non-continuous processes) at the molecular level. To investigate the solid-state stability of the saSMSDs, PXRD was utilized to show the morphological changes with time [45]. Taken together, our results highlight the advantages of the HME continuous manufacturing process by comparing short-term immediate drug release profiles and mechanisms of saSMSDs prepared using HME and two non-continuous amorphization techniques during individual stages of the short-term immediate drug release process.

## 2. Materials and Methods

### 2.1. Materials

Nitrendipine (NIT), a model BCS II drug, was supplied by Kangbaotai Fine Chemicals Co., Ltd., Wuhan, China at a purity of >99%. Polyvinyl caprolactam-polyvinyl acetate–polyethylene glycol graft copolymer (i.e., Soluplus^®^) was generously donated by BASF company (Ludwigshafen, Germany). NIT and polymer chemical structures and other detailed information are shown in Figure 1. All other chemicals and reagents used in this work were of analytical grade (or equivalent) and were purchased commercially.

### 2.2. Preparation of saSMSD/NIT Systems for Short-Term Immediate Release

#### 2.2.1. Preparation of saSMSD/NIT^C^_HME_ Using HME as a Continuous (C) Process

A hot melt extruder equipped with a feeding apparatus was employed to prepare saSmSD/NIT*^C^_HME_* based on a 1:4 (*w*/*w*) ratio of drug:polymer (Soluplus^®^) that was used to generate an initial physical mixture through uniform mixing. Next, a co-rotating twin-screw extruder (Pharma 11; Thermo Fisher Co., Shanghai, China) was used to extrude the mixture using a screw rate of 70 rpm and extrusion temperature of 121 °C to generate saSmSD/NIT*^C^_HME_*. Extruded preparations were cooled and allowed to solidify at room temperature then extrudates were milled using a laboratory mixer. Finally, milled extrudate powders were passed through an 80-mesh sieve then powders were stored in an airtight container at room temperature until further use.

#### 2.2.2. Preparation of saSMSD/NIT^NC^_CP_ Using Co-Precipitation (CP) as a Non-Continuous (NC) Process

A physical mixture containing a drug:carrier ratio of 1:4 (*w*/*w*) was prepared by mixing NIT with Soluplus^®^ to homogeneity in advance followed by addition of methanol to dissolve the mixture. Next, the methanol was evaporated using a rotary vacuum evaporator (RE520S, Yarong, Shanghai, China) maintained at a temperature of 40 °C and operating at a speed of 50 rpm. After the methanol was completely removed via evaporation, the crude co-precipitated sample preparations were transferred to a vacuum dryer for 24 h then the crude dry powders were passed through 80-mesh sieves and stored in a desiccator until further evaluation.

#### 2.2.3. Preparation of saSMSD/NIT^NC^_FQC_ Using Fusion/Quench Cooling (FQC) as a Non-Continuous (NC) Process

A non-continuous process, a fusion/quench cooling method (FQC), was employed to prepare saSMSD/NIT*^NC^_FQC_* from an initial physical mixture as a 1:4 (*w*/*w*) ratio of NIT to Soluplus^®^. Next, a fixed amount of physical mixture was transferred to a porcelain crucible and exposed to microwave irradiation to produce homogeneous fused liquids using a domestic microwave oven (P70F23P-G5(SO), Galz, Guangzhou, China). Thereafter, the fused liquids were rapidly quench-cooled to an extremely low temperature (−196 °C) in liquid nitrogen then each frozen sample was transferred to a glass mortar and ground with a pestle. The resulting powders were passed through 80-mesh sieve to obtain particles of uniform size. Finally, saSMSD/NIT*^NC^_FQC_* powders were stored in a desiccator for future analysis.

#### 2.2.4. Preparation of NIT as Physical Mixture (PM) and Pure Amorphous (AM) Forms

NIT and self-micellizing Soluplus^®^ were mixed vigorously and uniformly based on a drug:polymer ratio of 1:4 (*w*/*w*) for over 10 min in plastic bags to yield a physical mixture (NIT_PM_). To generate pure amorphous NIT (NIT_AM_), crystalline NIT was transferred to preheated porcelain crucibles then the drug was heated in an external dimethicone bath until the initial point of complete fusion. Next, the fused solution was maintained in a homogenous liquid state for 5 min then the fused liquid was subjected to rapid quench-cooling using liquid nitrogen (−196 °C). To prevent moisture absorption, the resulting chilled solids were immediately stored in a desiccator and kept at room temperature. Next, the dried samples were pulverized in a mortar using a pestle then the resulting pure amorphous NIT powder (NIT_AM_) was passed through a sieve (80-mesh) and stored in a desiccator until needed for analysis. Detailed abbreviations of various samples are listed in Table 1.

### 2.3. Scanning Electron Microscopy (SEM)

Scanning electron microscopy (SEM) (JSM-6490LV, JEOL, Tokyo, Japan) was employed to observe surface morphologies of various saSMSD/NIT systems and control preparations. The samples were prepared by affixing dried powder to a glass stub sample holder using double-sided adhesive tape. Next, samples were coated with gold to make them electrically conductive then images were captured using an accelerated voltage of 20.0 kV and analyzed for changes in polymorphic state.

### 2.4. Differential Scanning Calorimetry (DSC)

DSC measurements were carried out to evaluate the amorphous state of NIT in saSMSD/NIT systems using a differential scanning calorimeter (SDT-Q600, TA Instruments, Wakefield, MA, USA). A standard was established using indium for use in calibrating the sensitivity of the instrument. Approximate 5-mg samples were placed into an aluminum pan then nitrogen gas was applied as purge gas to maintain inertness of the DSC cell. The heating rate was 10 °C/min over a temperature range of 50 °C to 240 °C.

### 2.5. Powder X-ray Diffraction (PXRD)

An X-ray powder diffractometer (Rigaku, Tokyo, Japan) equipped with a Cu-Kα radiation (λ = 1.541 Å) source was used to analyze the amorphous state. The tube voltage and amperage were maintained at 40 kV and 40 mA, respectively. For conducting continuous determinations, XRPD data were collected within the 2*θ* range from 5° to 45°. Scanning speed and step size were set to 10°/min and 0.04°, respectively.

### 2.6. Measurement of Initial NIT Release Using Dissolution Assays

A ZRS-8G dissolution apparatus (Tianda, Tianjin, China) was used to determine sample NIT dissolution profiles. A saSMSD/NIT preparation containing 30 mg of NIT was accurately weighed and placed in a dissolution cup containing 900 mL medium such as PBS solution (pH 4.5, 6.8 and 7.5), 0.1 mol/L HCl or H_2_O. The dissolution test temperature was set to 37 ± 0.5 °C and the rotating speed of the paddle was set to 100 rpm. At given time intervals (5, 10, 15, 30, 60, 90, and 120 min), 5-mL aliquots were withdrawn and replaced with an equal volume of fresh medium then sample solutions were filtered through a 0.45-μm Millipore filter and assayed for NIT content using UV-visible spectroscopy at a wavelength of 350 nm. Each determination was performed in triplicate.

### 2.7. Evolved Release Stage “Spring-Parachute” Analysis

#### 2.7.1. Semi-Continuous Determinations

“Spring-parachute” profiles were obtained through collection of long-term (36 h) dissolution data from samples and controls. Sample equivalents of 30 mg NIT in saSMSD/NIT systems and controls were dissolved in 900 mL of PBS (pH 6.8) and placed within the dissolution apparatus. Next, an aliquot from each solution was removed at 0.5-h intervals throughout a period of 36 h (*n* = 1) then “spring-parachute” processes were assessed based on data collected through semi-continuous monitoring.

#### 2.7.2. Quantification

Aliquots of samples obtained in Section 2.7.1 were filtered through 0.45-μm Millipore filters then NIT contents in filtrates were measured using UV–visible spectroscopy at a wavelength of 350 nm according to assay methods mentioned above. Data were obtained to determine “spring and parachute” curves then corresponding parameters were calculated or simulated to quantify the process.

### 2.8. Solubility Measurements

Equilibrium solubilities were measured in 0.1 mol/L HCl, PBS (pH 4.5, 6.8, 7.5), and H_2_O. Excess amounts of samples were added into glass vials containing 50 mL of the abovementioned solutions. After vials were sealed, they were placed on an oscillating shaker at 37 °C for 24 h and 48 h. Thereafter, samples were filtered through a 0.45-μm Millipore filter and analyzed using UV/VIS spectrometry (UV-1800, Shimadzu, KYOTO, Japan) conducted at a *λ*_max_ of 350 nm. Each sample was analyzed in triplicate.

### 2.9. Long-Term Amorphous State Stability Assessments Conducted Using PXRD

Powdered preparations of saSMSD/NIT and other control groups were placed into sealed glass bottles under controlled conditions (temperature: 25 °C, dry conditions). Long-term stability of the amorphous state was determined after 1 month of storage based on crystallization characteristics as assessed using PXRD. Any peaks appearing in the PXRD spectra revealed the occurrence of crystallization that indicated instability of the amorphous state.

### 2.10. Phase Solubility Measurements

Phase solubility was measured in triplicate based on a previously reported method [25,46]. A series of aqueous Soluplus^®^ solutions in PBS (pH 6.8) that contained increasing Soluplus^®^ concentrations (0–8 mg/mL) were prepared then an excess amount of NIT added to these Soluplus^®^ solutions in flasks. Next, flasks were maintained in a sealed state and shaken for over 24 h thermostatically at 37 °C in a stable air bath. A 0.45-mm filter was used to filter the samples then diluted filtrates were analyzed for NIT concentrations spectrophotometrically at a wavelength of 350 nm according to methods described in Section 2.6.

### 2.11. Fourier Transform-Infrared Spectroscopy Analysis

An Agilent Cary 660 FT-IR Spectrometer was used to evaluate FT-IR behavior in order to study the interaction between NIT and polymer in saSMSD/NIT systems. Dry potassium bromide/sample mixture was prepared by mixing potassium bromide with 2~3 mg of each sample then the mixture was compressed into a pellet using a powder-compressing instrument. The FT-IR spectrum was analyzed within the wavelength range of 400 to 4000 cm^−1^ using a resolution set to 1 cm^−1^.

### 2.12. Crystallization Inhibition from a Supersaturated State

A crystallization simulation experiment was established to explore the ability of Soluplus^®^ to inhibit crystallization of NIT from a supersaturated state. Soluplus^®^ was pre-dissolved in 900 mL of PBS solution (pH 6.8), yielding a series of solutions of concentrations including 0, 0.05, 0.1, 0.15, and 0.35 mg/mL. Crystallization inhibition experimental conditions were controlled for temperature (37 ± 0.2 °C) and stirring rate (100 rpm). Concentrated NIT-methanol solution was rapidly injected into dissolution media to generate an initial NIT concentration of 33.3 mg/mL. Next, 5-mL aliquots of samples were taken at intervals of 15, 30, 60, 90, 120, 150, 180, 210, and 240 min then aliquots were filtered through 0.45-mm filters. Thereafter, NIT concentrations were measured using the aforementioned UV-VIS method (Section 2.6).

### 2.13. Data Analysis

Statistical analysis was carried out using an unpaired Student’s *t*-test. Data were indicated as the mean ± standard deviation (SD). Each statistical experiment was tested in triplicate (*n* = 3), with *p* < 0.05 and *p* < 0.01 defined as significant.

## 3. Results and Discussion

### 3.1. Preparation and Characterization of saSMSD/NIT Systems Prepared Using Continuous and Non-Continuous Processes

Surface micromorphologies of saSMSD/NIT preparations and control groups reflecting particle shapes, sizes, and microstructures as determined using SEM, are presented in Figure 2. Unprocessed crystalline NIT displayed a long ribbon-shaped appearance (Figure 2A), with a relatively regular arrangement (see magnified image in Figure 2B), indicating the presence of crystalline NIT, a water-insoluble form of the drug. Meanwhile, the polymer Soluplus^®^ when present alone contained particles with spheroidal shapes (Figure 2D), which itself is naturally amorphous. Importantly, physical mixtures of crystalline NIT and polymer exhibited similar physical micromorphological characteristics (Figure 2C) as those observed for crystalline NIT. By contrast, once NIT amorphization occurred, the micromorphology of the drug changed markedly, as reflected by the presence of particles with wrinkled and irregular shapes, resembling pure amorphous NIT particles (Figure 2E) that resulted from the effects of amorphization-inducing heat treatment. Meanwhile, non-continuous CP effects on saSMSD/NIT^NC^_CP_ systems led to changes in particle appearance from rough-surfaced ribbon-shaped microstructures to relatively smoother-surfaced block-shaped microstructures (Figure 2F), while non-continuous FQC and continuous HME techniques led to block-shaped microstructures with rougher surfaces (Figure 2G,H), resulting mainly from the effects of thermal treatment.

Another assessment tool, DSC curves, which can reflect the thermal behaviors of saSMSD/NIT systems and controls, can also provide detailed energy information to describe constituent drugs [47]. Thus, DSC analysis was conducted here to assess NIT phase transformation during the formation of saSMSD/NIT systems, crystalline NIT, and pure amorphous NIT (NIT_AM_). Figure 3 shows DSC thermographs of various samples collected within the temperature range of 50 to 240 °C. Notably, an obvious endothermic inverted peak (i.e., melting point) appeared in the crystalline NIT sample at a temperature of about 159.94 °C. By contrast, no endothermic peaks were detected for amorphous self-micellizing Soluplus^®^ (Figure 3C), while the physical mixture of NIT and polymer exhibited an endothermic inverted peak at 176.85 °C (Figure 3B). Sometimes melting point shifts are observed for physical mixtures (as compared to constituents) due to differential molecular miscibility within fused physical mixture solutions. However, pure amorphous NIT and saSMSD/NIT system curves lacked detectable melting point peaks, thus indicating that saSMSD/NIT^C^_HME_, saSMSD/NIT^NC^_FQC_, and saSMSD/NIT^NC^_CP_ were present in amorphous states regardless of whether they were fabricated using continuous or non-continuous processes.

As an additional type of analysis, PXRD used with DSC thermographs can facilitate verification of the amorphous state by determining whether individual crystalline substances are transformed into amorphous states [45,48]. As shown in Figure 4, crystalline NIT DSC thermographs exhibited obvious crystalline peaks at 10°, 11.32°, 13.12°, 13.6°, 23.76°, 24.36°, 25.96°, and 27.44° that reflected its crystalline structure. Interestingly, although the main crystalline peaks were still detected after physical mixing (at 10.04°, 11.32°, 13.08°, 13.64°, 23.68°, 24.4°, 26°, and 27.56°), the peaks exhibited reduced intensity. These results may be explained by the fact that the spatial confinement effect of the porous network structure of Soluplus^®^ significantly reduced the extent of NIT crystallinity, such that NIT assumed only a microcrystalline state within the porous Soluplus^®^ network. Further analysis of diffraction spectra indicated that Soluplus^®^ lacked crystalline peaks, a result in accordance with DSC-based results obtained for amorphous materials, including pure amorphous NIT. Importantly, a 100% amorphous state was observed for all types of saSMSD/NIT preparations (saSMSD/NIT^C^_HME_, saSMSD/NIT^NC^_FQC_, saSMSD/NIT^NC^_CP_). Thus, from these observations, it is readily apparent that saSMSD/NIT systems assumed completely amorphous states regardless of whether they were prepared using continuous or non-continuous processes.

### 3.2. SaSMSD/NIT Characteristics: Extended Dissolution, Spring-Parachute and Solubility

The absorption rate of water-insoluble NIT has been shown to be limited by its poor dissolution [49], as the main reason for the low bioavailability of the drug, as reflected by its initial immediate release profile. Thus, dissolution as a critical indicator of satisfactory absorption and improved therapeutic effect should be evaluated in various media in vitro to mimic pH levels within various in vivo environments [50]. Figure 5 depicts dissolution curves of saSMSD/NIT systems and control groups in various media for 0.1 M HCl, three PBS solutions (pH 4.5, 6.8, and 7.4), and H_2_O. Dissolution measurements conducted to predict in vivo absorption performance of saSMSD/INT systems in 0.1 M HCl solution (pH 1) were carried out to simulate gastric fluid, as shown in Figure 5A, with the results revealing low-level dissolution of approximately 0~2 μg/mL for crystalline NIT and the NIT-polymer physical mixture during a 120-min time period. By contrast, the single amorphous drug (NIT_AM_) exhibited markedly improved dissolution relative to that of crystalline NIT, suggesting amorphization provided a dissolution advantage that was most markedly apparent in the gastric environment as compared to neutral or intestinal environments. Meanwhile, saSMSD/NIT^C^_HME_, saSMSD/NIT^NC^_FQC_, and saSMSD/NIT^NC^_CP_ systems exhibited significantly improved dissolution behaviors and 3~4-fold higher concentrations than those observed for NIT_AM_. These improvements may have been due to effects associated with self-micellizing Soluplus^®^, which can form micelles in aqueous solution that may have been largely responsible for increased dissolution of saSMSD/NIT systems as compared to NIT_AM_ [51], although dissolution increases varied. More specifically, saSMSD/NIT^C^_HME_ induced extended dissolution as compared to saSMSD/NIT^NC^_FQC_ and saSMSD/NIT^NC^_CP_ during a period of 20–100 min, thus demonstrating the superior effect of the saSMSD/NIT system generated using a continuous process, while saSMSD/NIT^NC^_CP_ generated using a non-continuous process exhibited the lowest dissolution of all of the saSMSD/NIT systems. However, in PBS solution of pH 4.5 (as shown in Figure 5B), dissolution of NIT_AM_ and the physical mixture were similar and extremely low (<1 μg/mL) at 120 min as compared to markedly increased dissolution values observed for saSMSD/NIT systems, with extension of dissolution most apparent for saSMSD/NIT^C^_HME_ relative to saSMSD/NIT^NC^_FQC_ and saSMSD/NIT^NC^_CP_. Moreover, dissolution behaviors of saSMSD/NIT systems in PBS at pH 6.8, as shown in Figure 5C, were similar to dissolution behaviors observed for systems in PBS at pH 4.5 (with the exception of saSMSD/NIT^NC^_FQC_, which exhibited lower dissolution values as compared to values for saSMSD/NIT^C^_HME_ and saSMSD/NIT^NC^_CP_), while opposite trends were observed in PBS (pH 7.4) (see Figure 5D). It should be noted that regardless of the dissolution medium used, the HME-induced continuous process almost always achieves extended dissolution. In addition, measurements of dissolution rates (Figure 6) based on data of dissolution measured at 30 min, 60 min, and 120 min revealed generally higher dissolution rates at 30 min than at the other two timepoints, suggesting that the most rapid dissolution occurred during the initial 30 min for all saSMSD/NIT systems. Notably, the highest dissolution rate of ~0.12–0.18 μg/(mL∙min) was observed for the saSMSD/NIT^C^_HME_ system, while the lowest dissolution rates were observed for NIT, NIT_AM_, and NIT_PM_ systems. Self-micellizing polymer Soluplus^®^ can form micelles in aqueous solution [28] and allocate NIT molecules inside, which is helpful to maintain the stabilization of the supersaturation state that contributes to the dissolution improvement of all saSMSD systems. Moreover, enhanced crystallization inhibition capacity by self-micellizing polymer Soluplus^®^ against NIT (see below Section 3.3) and molecular interaction (see below Section 3.3) are also both favorable to stabilizing the amorphism of the NIT molecule, which is also beneficial to the dissolution improvement in saSMSD systems. Taken together, these results suggest that the HME continuous process provided a unique advantage by extending the dissolution duration and increasing drug dissolution rates of a formed supersaturable amorphous SMSD system.

Supersaturation is important for increasing drug absorption, with supersaturation concentration varying for different supersaturable formulations. Importantly, the supersaturated state is unstable and tends to revert to a more stable crystalline state. To extend the supersaturated state in order to benefit from it, it is critical to elevate the “spring” height and delay the crystallization process in order to prolong the “parachute” period. Thus, monitoring of the “spring-parachute” process is essential for evaluating the abilities of supersaturable formulations (e.g., saSMSDs) to maintain supersaturation throughout the entire process. In this work, we dynamically monitored the “spring-parachute” process using a representative medium of PBS at pH 6.8 in a semi-continuous manner for 36 h, with results obtained for saSMSD systems and controls shown in Figure 7. Due to the fact that NIT and NIT_PM_ were not subjected to any formulation process, no evidence of supersaturation was found for either drug form, as supported by their observed concentrations throughout the entire process of close to 1~2 μg/mL. By contrast, amorphization of pure drug (NIT_AM_) induced a change in and elevation of the spring-parachute process, with maximum concentration levels reaching about 7 μg/mL. This result thus highlights the advantage of amorphization, such that once NIT was incorporated into a saSMSD system, the spring-parachute process was moved up to a higher level than that of NIT_AM_. Notably, saSMSD/NIT^C^_HME_ exhibited the greatest extension of the spring-parachute process of the saSMSD/NIT systems, with the maximum NIT concentration reaching 9 μg/mL, while saSMSD/NIT^NC^_FQC_ and saSMSD/NIT^NC^_CP_ were associated with slightly lower spring-parachute processes, as reflected by NIT concentrations of 6~8 μg/mL. However, amorphization did not always trigger initiation of the spring-parachute process due to variations in the drug-polymer ratio, type of medium, and environmental temperature [52]. Nevertheless, maintenance of supersaturation, which is important for prolonging absorption of supersaturable formulations, acts to delay the crystallization process. In these self-micellizing systems, the existence of an amorphous structure has been verified by Figure 3; Figure 4, and is beneficial in elevating the “spring-parachute” process. The highest “spring-parachute” process caused by HME might be attributed to the formed advantageous amorphism due to excellent molecular miscibility via relatively complicated steps during the extrusion process [35]. From the results shown in Figure 7, during the 36-h assay period, almost no apparent parachute-based decline in crystallization was observed, suggesting that all saSMSD systems, regardless of whether they were produced by continuous or non-continuous processes, could maintain supersaturation, or inhibit crystallization to prolong crystallization time.

Induction of supersaturation is expected to elevate solubility, which is critical for enhancing drug absorption in the gastrointestinal tract (GT). Therefore, it is important to understand mechanisms affecting the performance of supersaturable formulations for elevating solubility using various representative media to simulate conditions in various regions of the GT, as shown in Figure 8; Figure 9 for representative media at 24 h and 48 h, respectively. Importantly, NIT and NIT_PM_ exhibited extremely low solubility <2 μg/mL) in all media, while amorphization of NIT led to enhancement of solubility, although this effect was limited due to the absence of crystallization inhibitors. Meanwhile, solubility enhancement was weaker in more acidic media (0.1 M HCl) as compared to that of near-neutral media. Ultimately, all saSMSD/NIT systems exhibited significantly elevated solubility to differing degrees, with the highest solubility observed for saSMSD/NIT^C^_HME_ as compared to solubilities of saSMSD/NIT^NC^_FQC_ and saSMSD/NIT^NC^_CP._ With regard to media effects, the solubility of saSMSD/NIT^C^_HME_ was lower in 0.1 M HCl than in other media, while saSMSD/NIT^NC^_FQC_ exhibited greater solubility enhancement in most media (except H_2_O) as compared with saSMSD/NIT^NC^_CP_, which exhibited the opposite trends. Moreover, solubility profiles at 48 h (Figure 9) were similar to those at 24 h (Figure 8). Based on all of these results, NIT solubility in various media (except for H_2_O) was ranked as follows: saSMSD/NIT^C^_HME_ > saSMSD/NIT^NC^_FQC_ > saSMSD/NIT^NC^_CP_ > NIT_AM_ > NIT_PM_ > NIT.

### 3.3. Stability Profiles and Underlying Molecular Mechanisms

The greatest challenge to preventing the maintenance of the physical stability of supersaturable formulations relates to the instability of the supersaturable form either during the liquid dissolution period or during the solid storage stage. PXRD provides a convenient approach for assessing the stability of an amorphous state of a supersaturable formulation with time. Here, various saSMSD/NIT and control group samples were stored under defined conditions for one month. As shown in Figure 10, after 1 month of storage, the amorphous drug NIT_AM_ exhibited a significant phase transition change, as reflected by the main crystalline peaks of NIT_AM_ detected at 9.68°, 11.36°, 12.92°, 13.84°, 23.96°, 24.32°, 25.92°, and 26.96°. This observation suggested that the supersaturable state was unstable and tended toward crystallization, resulting in the formation of more stable crystalline forms in the solid environment. By contrast, saSMSD/NIT^C^_HME_, saSMSD/NIT^NC^_FQC_, and saSMSD/NIT^NC^_CP_ exhibited markedly improved physical stability, as evidenced by a lack of PXRD crystal peaks after 1 month of storage. Taken together, these results indicated that saSMSD/NIT systems generated using either continuous or non-continuous processes exhibited markedly improved physical stability in liquid and solid environments as compared to those of NIT_AM_.

Exploration of underlying molecular mechanisms is essential for understanding how improvement in dissolution properties is achieved when using saSMSD/NIT systems, with roles of intermolecular interactions possibly responsible for observed improvements. To identify intermolecular interactions (e.g., hydrogen bonding interactions), FT-IR spectroscopy is a well-established approach for exploring drug-polymer interactions occurring in saSMSD/NIT systems. Here results obtained using FT-IR are shown in Figure 11 for saSMSD/NIT systems and controls. Pure NIT exhibited four strong peaks, including peaks corresponding to N–H stretch (3315 cm^−1^), a saturated C-H characteristic absorption peak (2976 cm^−1^), and two asymmetric C=O stretching vibration absorption peaks associated with ester groups (1701 cm^−1^ and 1651 cm^−1^). Meanwhile, the Soluplus^®^ spectrum contained four characteristic peaks, including one O–H stretching peak at 3446 cm^−1^, a caprolactam peak (1635 cm^−1^), and an ether C-O-C peak (1242 cm^−1^), which was similar to previous literature [53]. For the physical mixture, the spectrum reflected the superposition of spectra of the pure ingredients, with attenuation of NIT peaks observed as the only significant difference found between the spectra of the physical mixture and the spectra of the pure components. However, FT-IR spectra of saSMSD/NIT systems exhibited several significant differences as compared to the spectrum of the physical mixture, whereby the NIT peak almost disappeared while most of the peaks corresponding to free Soluplus^®^ were still visible, suggesting that some type of interaction, probably a hydrogen bonding interaction, occurred between NIT and Soluplus^®^ in the complex. The molecular interaction can also be observed in other publications [54]. At the same time, the saSMSD/NIT systems spectra did not reveal any new peaks, thus indicating no other chemical bonds were created during saSMSD/NIT system formation.

It is important to assess the capacity of the self-micellizing Soluplus^®^ polymer for suppressing crystallization of NIT from a supersaturated state since the polymer can inhibit the growth of crystal nuclei to critically influence the stability of the supersaturable state both in liquid and solid systems. Toward this end, a simulated crystallization suppression experiment (see Figure 12) was used to assess the crystallization inhibition ability of self-micellizing Soluplus^®^ polymer against supersaturated NIT in a liquid environment, with the results expressed as the crystallization half-time (Table 2). In order to determine crystallization half-time values, a concentrated NIT solution in methanol was rapidly infused into the dissolution medium, and its concentration was determined over a 240-min period of time. Once the infusion procedure was initiated, crystallization of the supersaturated drug in the system occurred without further assistance, for an initial NIT concentration of 33.33 μg/mL. In the absence of self-micellizing material, NIT crystallization from the supersaturated state occurred with greatest rapidity and exhibited a crystallization half-time of about 30 min (Figure 12A) and the highest crystallization rate of almost 0.12 μg/(mL∙min). By contrast, the addition of pre-dissolved Soluplus^®^ polymer significantly prolonged the crystallization half-time, with the crystallization inhibitory effect becoming stronger as the concentration of pre-dissolved Soluplus^®^ was increased. At the highest Soluplus^®^ concentration of 0.35 mg/mL, the strongest growth inhibition of crystal nuclei was observed that corresponded to the longest half-time observed (170 min) and the slowest crystallization rate of ~0.09 μg/(mL∙min) (Figure 12B). Thus, these results suggest that crystallization inhibition by self-micellizing Soluplus^®^ polymer was a main contributor to the extended “spring-parachute” process and the observed increases in dissolution/solubility effects.

Self-micellizing Soluplus^®^ has been observed to exert excellent solubilization effects on many BCS II drugs while also serving as a matrix material and crystallization inhibitor, properties that stem from its amphiphilic properties and its ability to form micelles in aqueous solution. However, it is unknown whether Soluplus^®^ can solubilize NIT, prompting this study. Here, the NIT solubilization profile was evaluated based on phase solubility (as shown in Figure 13) and Gibbs free energy (summarized in Table 3). Phase solubility was measured by adding excess NIT into a series of Soluplus^®^/PBS (pH 6.8) solutions containing various pre-dissolved polymer concentrations. From Figure 13, it can be seen that in the absence of Soluplus^®^ the solubility of NIT was ~1.42 μg/mL, with NIT phase solubility increasing significantly (*p* < 0.01) with increasing concentration of pre-dissolved Soluplus^®^. Notably, a high Soluplus^®^ concentration of 0.8 mg/mL supported the greatest NIT phase solubility observed (~22.49 μg/mL), a solubility level that was ~16-fold higher than the corresponding value obtained without Soluplus^®^. Furthermore, from the curve, it can be seen that the apparent solubility of NIT increased linearly as a function of Soluplus^®^ concentration (correlation coefficient: 0.9506), resulting in an AL-type phase solubility curve, thus suggesting that the soluble NIT:Soluplus^®^ complex formed in a 1:1 stoichiometric ratio, as previously reported by Highuchi and Connors [55]. Importantly, self-micellizing Soluplus^®^ exhibited a marked ability to support NIT solubilization, since negative values for $\Delta G_{tr}^{o}$ were obtained for NIT in the presence of Soluplus^®^ that ranged from −3.27 to −6.84 KJ/mol and decreased with increasing carrier concentration. The Gibbs free energy is calculated from Equation (1). This result revealed that the process of NIT transfer from PBS solution to carrier solution became more favorable and spontaneous at higher carrier concentrations, thus demonstrating that Soluplus^®^ could act as a crystallization inhibitor that also solubilized poorly water-soluble NIT.
(1)$$\Delta G_{tr}^{o}=-2.303RT\cdot \log\frac{S_{o}}{S_{s}}$$

## 4. Conclusions

In this study, three saSMSD/NIT systems were prepared and amorphized using a HME continuous manufacturing production process and FQC and CP non-continuous manufacturing processes. By comparing these amorphization technologies, all three categories of manufacturing techniques were investigated based on an identical drug loading rate of 20% for all NIT-polymer combinations evaluated for each technique. Subsequently, results of SEM, DSC, and PXRD analyses all revealed that NIT was present in amorphous form in all three saSMSD/NIT systems prepared using the three different amorphization techniques. Furthermore, amorphous states of NIT in saSMSD/NIT systems prepared using FQC, CP, or HME processes were equally stable after 1 month of storage and exhibited greater stability than that observed for pure amorphous NIT. Ultimately, HME-triggered amorphization, a continuous manufacturing process, was found to be the most effective due to its ability to enhance solubility and extend the dissolution of saSMSD/NIT^C^_HME_ in different media, although effects varied with medium selection and assay time. Moreover, greater maintenance of the “spring-parachute” process by the saSMSD/NIT^C^_HME_ system was evident, as based on the AUC_spring-parachute_, C_max,_ and T_max_ values obtained throughout the process. To investigate molecular mechanisms underlying internal properties of saSMSD/NIT systems, FT-IR spectroscopy, simulated crystallization inhibition and phase solubility experiments were conducted. Importantly, FT-IR results revealed that the intermolecular interactions between NIT and Soluplus^®^ occurred only in saSMSD/NIT systems, suggesting that the generation of intermolecular hydrogen bonds was responsible for increased stability of amorphous NIT within saSMSD/NIT systems, as consistent with the dissolution, solubility, and dissolution rate results obtained here. Nevertheless, molecular interactions did not mechanistically explain the relatively greater stability of amorphous NIT in the saSMSD/NIT^C^_HME_ system. However, crystallization inhibition and half-time assays revealed that self-micellizing Soluplus^®^ effectively inhibited crystallization of NIT from a supersaturated state by stabilizing saSMSD systems to delay crystallization. Furthermore, $\Delta G_{tr}^{o}$ determinations revealed that Soluplus^®^ possessed a high Gibbs free energy of transfer that enabled the polymer to solubilize NIT in a spontaneous manner. Taken together, the results of this study have enhanced our understanding of the differences between saSMSD systems generated using continuous versus non-continuous manufacturing processes. Nonetheless, our understanding of deeper mechanisms underlying improved HME-triggered amorphization and performance of saSMSD/NIT systems for extending the short-term immediate release of NIT must await further investigations. In the meantime, results obtained here provide valuable information to guide the development of BCS II drug-forming saSMSD systems for future use in clinical settings.

## Figures and Tables

**Figure 1 pharmaceutics-14-00765-f001:**
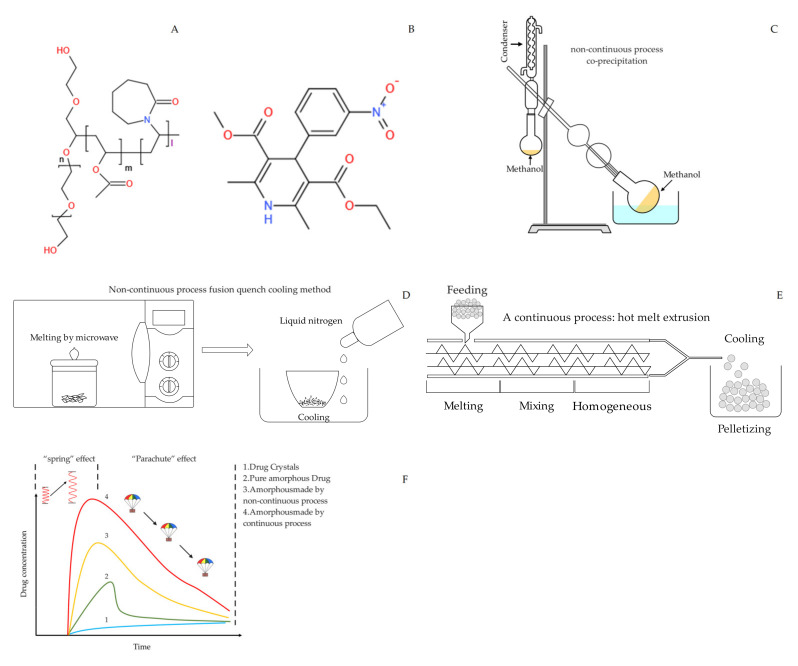
Chemical structures of drug and polymer (**A**) Soluplus^®^; (**B**) NIT. Preparation process (**C**) Schematic diagram of co-precipitation process; (**D**) Schematic diagram of fusion/quench cooling process; (**E**) Schematic diagram of hot melt extrusion progress. (**F**) Illustration of the spring-parachute process associated with saSMSDs made by continuous and non-continuous preparation processes and other control groups.

**Figure 2 pharmaceutics-14-00765-f002:**
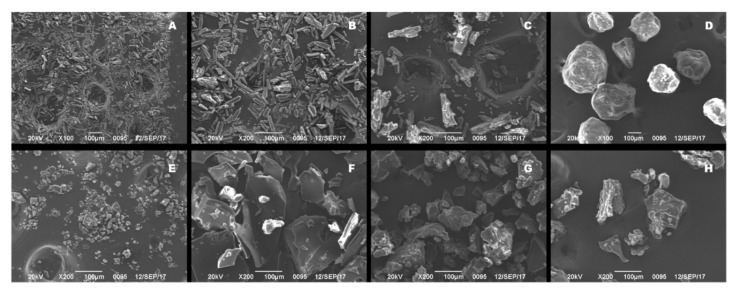
Analysis of scanning electron microscope on crystalline NIT ((**A**), 100×), crystalline NIT ((**B**), 200×), NIT_PM_ ((**C**), 200×), Soluplus^®^ ((**D**), 200×), NIT_AM_ ((**E**), 200×), saSMSD/*NIT^NC^_CP_* ((**F**), 200×), saSMSD/*NIT^NC^_FQC_* ((**G**), 200×), saSMSD/*NIT^C^_HME_* ((**H**), 200×).

**Figure 3 pharmaceutics-14-00765-f003:**
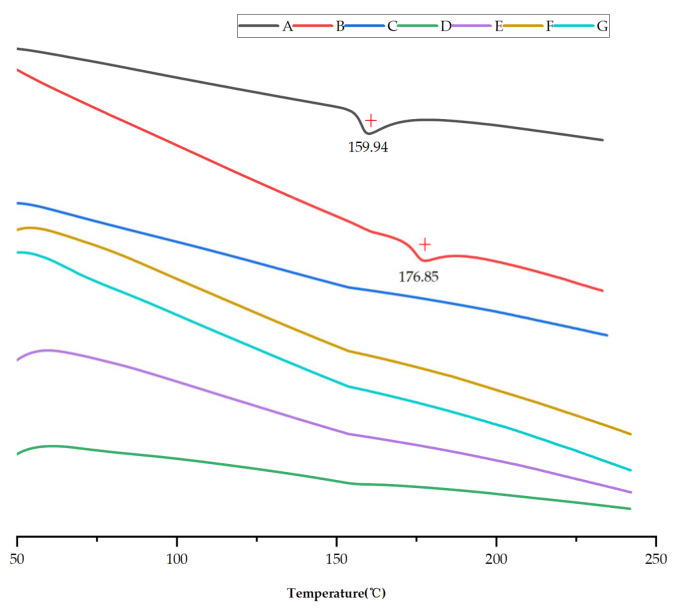
DSC curves of NIT (**A**), NIT_PM_ (**B**), Soluplus^®^ (**C**), NIT_AM_ ((**D**), 200×), saSMSD/*NIT^NC^_CP_* (**E**), saSMSD/*NIT^NC^_FQC_* (**F**), saSMSD/*NIT^C^_HME_* (**G**).

**Figure 4 pharmaceutics-14-00765-f004:**
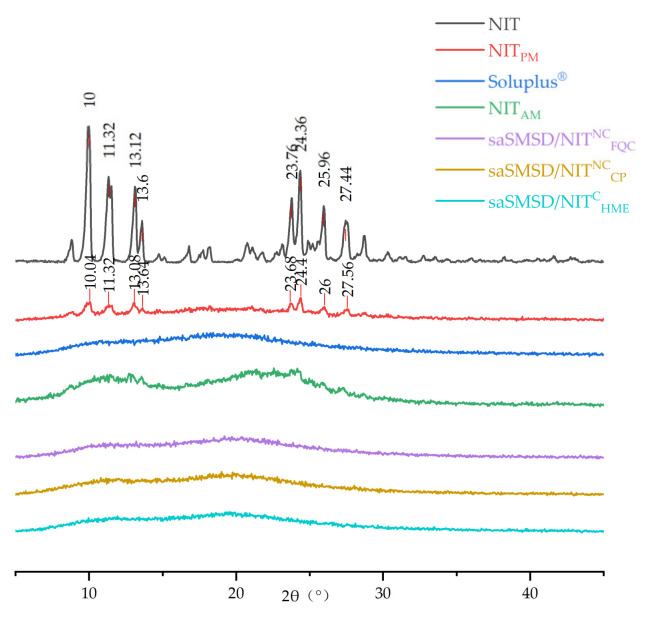
PXRD spectra of NIT, NIT_PM_, Soluplus^®^, NIT_AM_, saSMSD/*NIT^NC^_CP_*, saSMSD/*NIT^NC^_FQC_*, saSMSD/*NIT^C^_HME_*.

**Figure 5 pharmaceutics-14-00765-f005:**
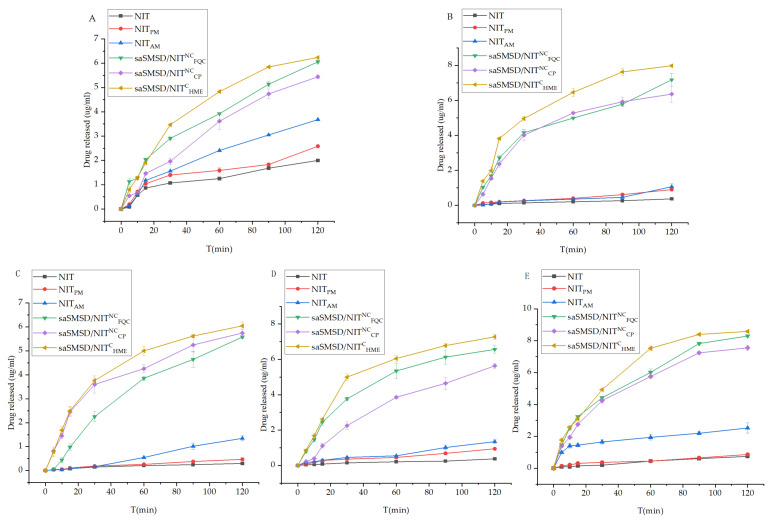
(**A**) Dissolution profiles of NIT, NIT_PM_, NIT_AM_, saSMSD/*NIT^NC^_FQC_*, saSMSD/*NIT^NC^_CP_*, saSMSD/*NIT^C^_HME_* in 0.1mol/L HCl; (**B**) Dissolution profiles of NIT, NIT_PM_, NIT_AM_, saSMSD/*NIT^NC^_FQC_*, saSMSD/*NIT^NC^_CP_*, saSMSD/*NIT^C^_HME_* in PBS (pH4.5); (**C**) Dissolution profiles of NIT, NIT_PM_, NIT_AM_, saSMSD/*NIT^NC^_FQC_*, saSMSD/*NIT^NC^_CP_*, saSMSD/*NIT^C^_HME_* in PBS (pH6.8); (**D**) Dissolution profiles of NIT, NIT_PM_, NIT_AM_, saSMSD/*NIT^NC^_FQC_*, saSMSD/*NIT^NC^_CP_*, saSMSD/*NIT^C^_HME_* in PBS (pH7.5); (**E**) Dissolution profiles of NIT, NIT_PM_, NIT_AM_, saSMSD/*NIT^NC^_FQC_*, saSMSD/*NIT^NC^_CP_*, saSMSD/*NIT^C^_HME_* in H_2_O. Standard deviation expressed by error bars (*n* = 3).

**Figure 6 pharmaceutics-14-00765-f006:**
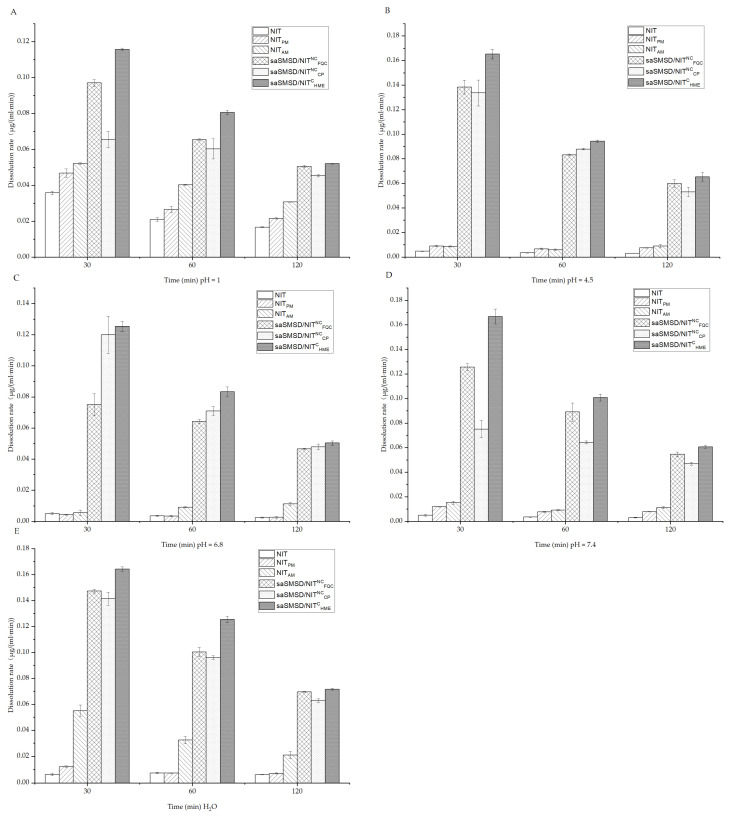
Dissolution rates calculated at 30, 60 and 120 min for NIT-related samples in 0.1 mol/L HCl (**A**); NIT-related samples in PBS (pH 4.5) (**B**); NIT-related samples in PBS (pH 6.8) (**C**); NIT-related samples in PBS (pH 7.5) (**D**); NIT-related samples in H_2_O (**E**). Standard deviations are expressed as error bars (*n* = 3).

**Figure 7 pharmaceutics-14-00765-f007:**
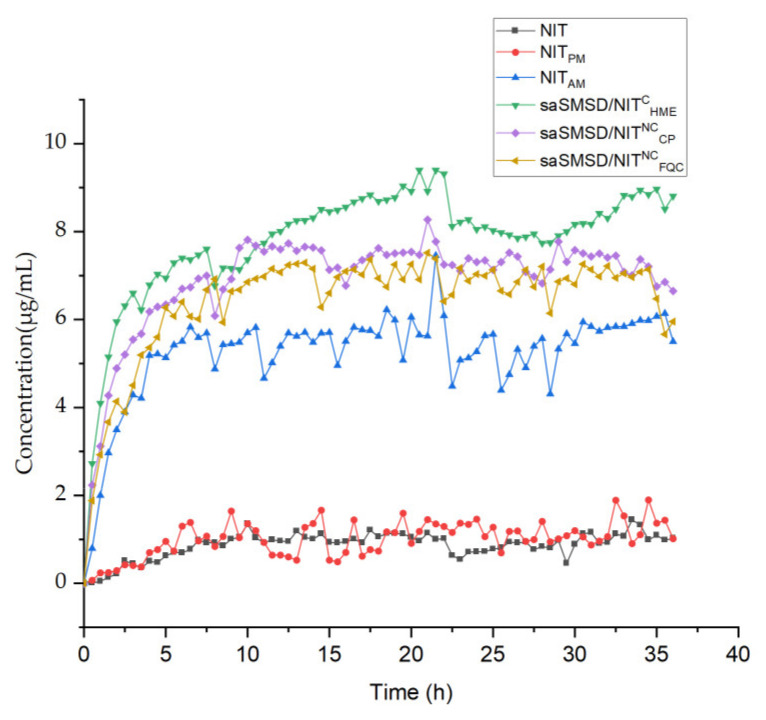
“Spring-parachute” profile of NIT-related samplesincluding NIT, NIT_PM_, NIT_AM_, saSMSD/*NIT^NC^_FQC_*, saSMSD/*NIT^NC^_CP_*, saSMSD/*NIT^C^_HME_* during 36 h at the equivalence of 30 mg NIT in PBS (pH 6.8) using dissolution apparatus.

**Figure 8 pharmaceutics-14-00765-f008:**
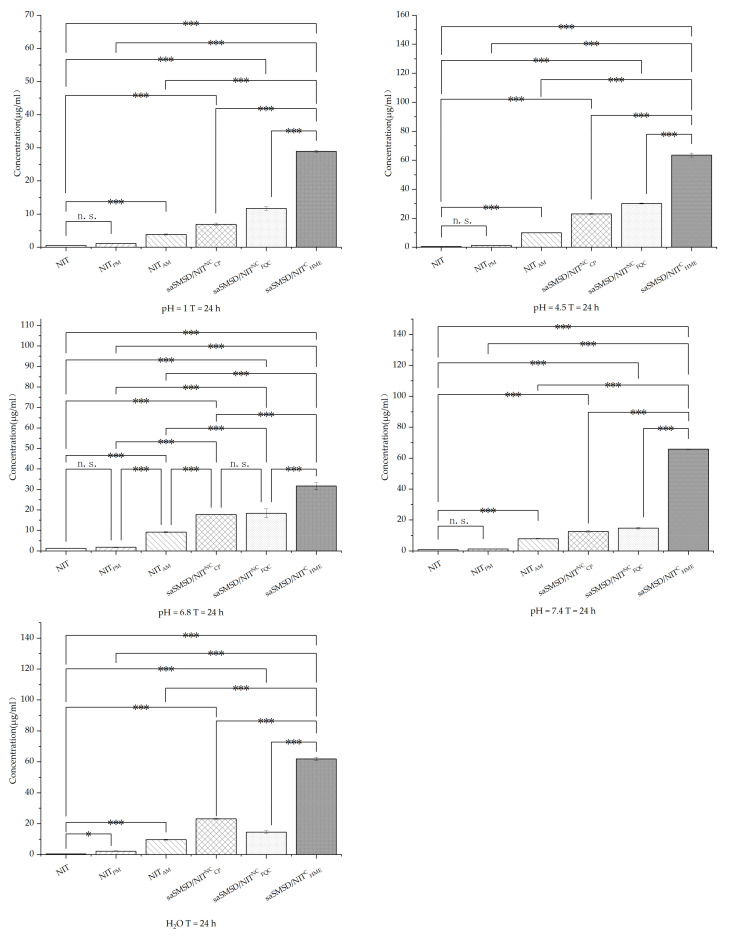
Solubility of samples in different solutions (H_2_O, 0.1 mol/L HCl and PBS (pH 4.5, 6.8, 7.5)) at 24 h; * *p* < 0.05, *** *p* < 0.001, n. s., no significant difference. Standard deviation expressed by error bars (*n* = 3).

**Figure 9 pharmaceutics-14-00765-f009:**
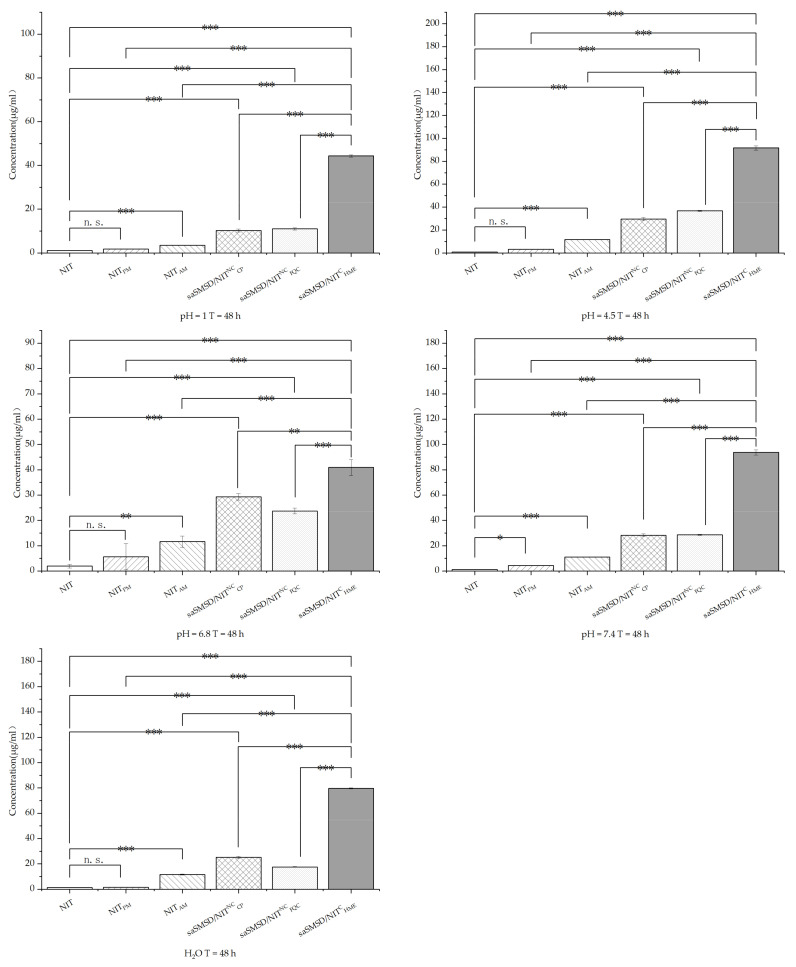
Solubility of samples in different solutions (H_2_O, 0.1 mol/L HCl and PBS (pH 4.5, 6.8, 7.5)) at 48 h; * *p* < 0.05, ** *p* < 0.01, *** *p* < 0.001, n. s., no significant difference. Standard deviation expressed by error bars (*n* = 3).

**Figure 10 pharmaceutics-14-00765-f010:**
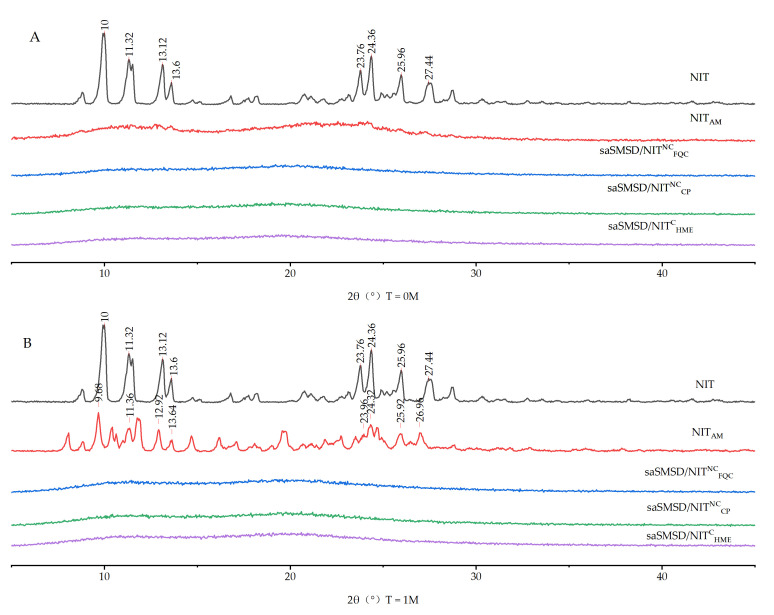
Stability Powder X-ray diffractogram of NIT, NIT_AM_, saSMSD/*NIT^NC^_FQC_*, saSMSD/*NIT^NC^_CP_*, saSMSD/*NIT^C^_HME_* at 0 month (**A**), 1 month (**B**).

**Figure 11 pharmaceutics-14-00765-f011:**
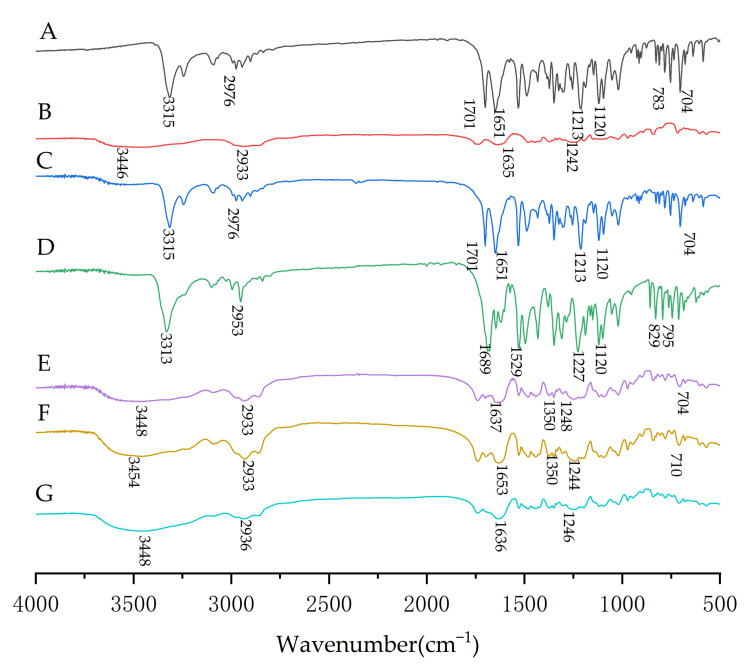
FTIR spectra of samples including NIT (**A**), Soluplus^®^ (**B**), NIT_PM_ (**C**), NIT_AM_ (**D**), saSMSD/*NIT^NC^_FQC_* (**E**), saSMSD/*NIT^NC^_CP_* (**F**), saSMSD/*NIT^C^_HME_* (**G**).

**Figure 12 pharmaceutics-14-00765-f012:**
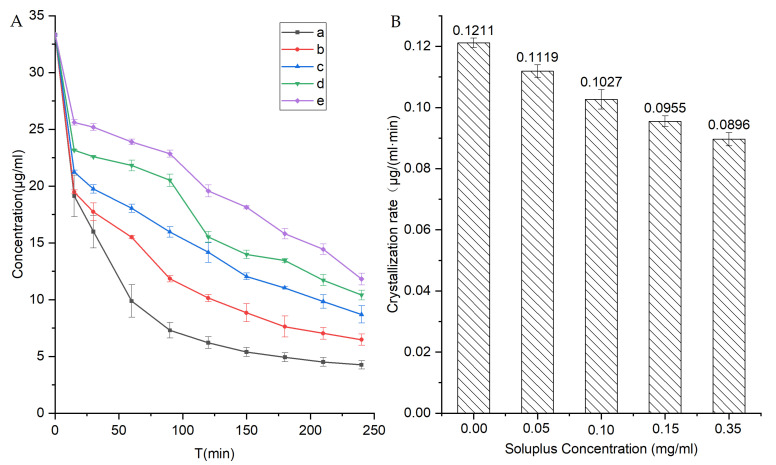
(**A**) Inhibition effects of Soluplus^®^ on the recrystallization of NIT from a supersaturated concentrated solution (33.33 μg/mL) at 37 °C in PBS (pH 6.8) (a) and mixed with Soluplus^®^ at different concentrations of 0.05 (b), 0.1 (c), 0.15 (d), or 0.35 mg/mL (e). (**B**) The relationship between crystallization rate and Soluplus^®^ concentration. Standard deviations are expressed as error bars (*n* = 3).

**Figure 13 pharmaceutics-14-00765-f013:**
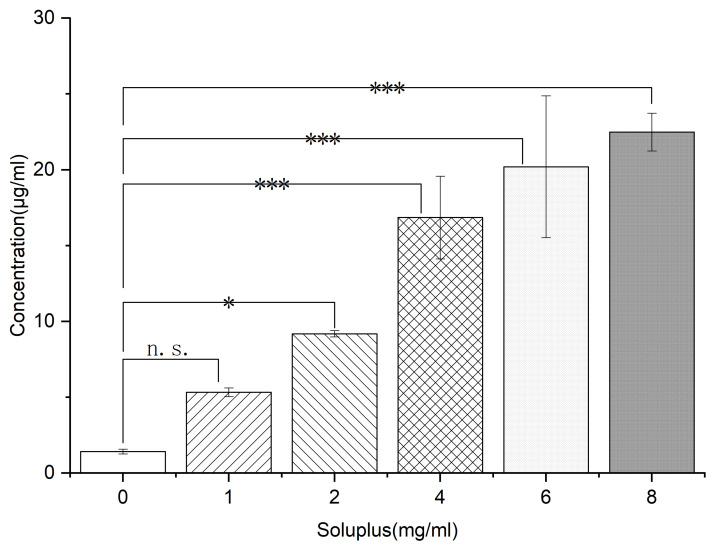
Phase solubility diagram of NIT in PBS (pH 6.8) mixed with pre-dissolved Soluplus^®^ polymers at different concentrations (from 0.1 to 0.8 mg/mL) at 37 °C. * *p* < 0.05, *** *p* < 0.001 n. s., no significant difference. Standard deviations are expressed as error bars (*n* = 3).

**Table 1 pharmaceutics-14-00765-t001:** Abbreviation of different samples in this research.

Samples	Abbreviation	Molecular Weight (Da)
Nitrendipine	NIT	360.36
Soluplus^®^	So	90,000–140,000
Physical mixtures	NIT_PM_	N/A
Pure amorphous nitrendipine	NIT_AM_	N/A
saSMSD made by HME technique	saSMSD/*NIT^C^_HME_*	N/A
saSMSD made by CP techniques	saSMSD/*NIT^NC^_CP_*	N/A
saSMSD made by FQC techniques	saSMSD/*NIT^NC^_FQC_*	N/A

**Table 2 pharmaceutics-14-00765-t002:** Half-time (T1/2) of recrystallization of crystalline NIT from a supersaturated concentrated solution (33.33 μg/mL) at 37 °C in a mixed solution of PBS (pH 6.8) without or with pre-dissolved Soluplus^®^ (0.05, 0.1, 0.15, or 0.35 mg/mL).

Concentration (mg/mL) of Pre-Dissolved Soluplus^®^	Initial NIT Concentration (μg/mL) in Supersaturated Solution	Half-Time (min) of NIT Crystallization from a Supersaturated Sate
0	33.33	30.55 ± 9.38
0.25	33.33	43.13 ± 4.45
0.1	33.33	80.51 ± 3.92
0.15	33.33	112.95 ± 1.98
0.35	33.33	170.22 ± 3.42

**Table 3 pharmaceutics-14-00765-t003:** The Gibbs free energy of transfer (ΔGtr°) of NIT in PBS (pH 6.8) with pre-dissolved Soluplus^®^ concentrations ranging from 0.1 to 0.8 mg/mL. Standard deviations are expressed as error bars (*n* = 3).

Concentration (mg/mL) of Soluplus^®^	Drug	ΔGtr° (KJ/mol)
0.1	NIT	−3.27 ± 0.13
0.3	NIT	−4.62 ± 0.06
0.5	NIT	−6.10 ± 0.41
0.7	NIT	−6.53 ± 0.62
0.9	NIT	−6.84 ± 0.14

## Data Availability

Data is available in the text.

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
