# Peer review of "Hot Melt Extrusion-Triggered Amorphization as a Continuous Process for Inducing Extended Supersaturable Drug Immediate-Release from saSMSDs Systems"

_pharmaceutics, 2022, doi:10.3390/pharmaceutics14040765_

Round 1

Reviewer 1 Report

The manuscript entitled “Hot melt extrusion-triggered amorphization as a continuous process for inducing extended supersaturable immediate-release of a BCS II drug from supersaturating amorphous self-micellizing solid dispersions” reports on an experimental study aimed at increasing the solubility in body fluids of the poorly water soluble drug nitrendipine. The strategy employed is the use of the amphiphilic polymer Soluplus for the formulation of the solid dispersions. The solid dispersion, prepared according various methods, HME included, were thoroughly characterized by  SEM, DSC, PXRD, solubility, FTIR spectroscopy.

In my opinion there are a few issues deserving attention:

- The title is lengthy. I find e.g. the one of ref. 38 more incisive

-Abtract The sentence at lines 23-25 is not explained in the text. I suggest to delete it

Introduction

-line 94: I do not find correct to define a process as “thermodynamically stable”, it might be defined reversible, if the relevant conditions hold.

-line 112: add a reference for the Noyes-Whitney equation, maybe the original work (THE RATE OF SOLUTION OF SOLID SUBSTANCES IN THEIR OWN SOLUTIONS. Arthur A. Noyes and Willis R. Whitney Journal of the American Chemical Society 1897 19 (12), 930-934 DOI: 10.1021/ja02086a003)

-line122: it is better to speck of amorphized drug solubility rather than “phase solubility”

-line 123: the Gibbs energy is related to the solubility, so it is the same.

Materials and methods

-line 244 Pay attention to the meaning of “phase” in thermodynamics

-line 245 Is ref 20 correct? It relies on fluorescence techniques.

-line 253 Were two different FTIR spectrometer used? Agilent and Thermo are two different producers.

Results

Since lines 277- 305 refer to literature, that text would be better placed in the Introduction

-lines 310-312 text not clear since Figure 2D reports the micrograph of pure Soloplus

-lines 502-519 The FTIR discussion would gain in efficacy taking in consideration works concerning the FTIR characterization of systems based on Soloplus and nifedipine, an analogue of nitrendipine, e.g. Kanika Sarpal, Eric J. Munson, Amorphous Solid Dispersions of Felodipine and Nifedipine with Soluplus®: Drug-Polymer Miscibility and Intermolecular Interactions, Journal of Pharmaceutical Sciences, Volume 110, Issue 4, 2021, 1457-1469, https://doi.org/10.1016/j.xphs.2020.12.022 and Mohammad A. Altamimi, Steven H. Neau, Investigation of the in vitro performance difference of drug-Soluplus® and drug-PEG 6000 dispersions when prepared using spray drying or lyophilization, Saudi Pharmaceutical Journal,  25, 3, 2017, 419-439, https://doi.org/10.1016/j.jsps.2016.09.013, where the Soloplus spectrum looks more defined. Maybe a better spectrum might be obtained by longer grinding the mixture with KBr.

Table 3 Report the drug concentration values also in the table, not only in fig 13. What does mean FEL? Indicate the units of ‘DELTA’Gtr°, kJ mol-1(as at line 573)

Write in the text the equation used for the calculation of ‘DELTA’Gtr° probably =-RTln([micellar solution]/[water]), like eq. (2) in Y. Kadam, U. Yerramilli, A. Bahadur, P. Bahadur, Micelles from PEO–PPO–PEO block copolymers as nanocontainers for solubilization of a poorly water soluble drug hydrochlorothiazide, Colloids and Surfaces B: Biointerfaces, 83, 1, 2011, 49-57, https://doi.org/10.1016/j.colsurfb.2010.10.041

Conclusions

line 560 state the process the Gibbs energy refers to: drug solubilization in Soluplus micellar solution

line 612 In thermodynamics the Gibbs energy is a criterion for the spontaneity of the process, it does not provide information about kinetics, thus delete instantaneous. Information on the rate of the solubilization process are provided by the experimental data of Figure 7.

Reviewer 2 Report

- The approach is interesting and the topic is appropriate for the journal.

  • The work has a very clear structure and all the sections are well written in a way that is easy to read and understand.
  • However, little modifications and improvements are needed to enhance the quality of the paper.

  • The paper is focused on hot melt extrusion-triggered amorphization as a continuous process for inducing extended supersaturable immediate-release of a BCS II drug  from supersaturating amorphous self-micellizing  solid dispersions, reporting very interesting results. In the “Introduction” section, the authors start to discuss about  some technical features related to the topic. Even though the authors already report some strategies in literature related to the topic and they especially focus on different aspects according to the aim of the work, I also suggest to BRIEFLY introduce the important role of some features related to rheological aspects of the extrusion process (i.e., viscosity as function of shear rate, material properties before and after the extrusion/injection, test for analyzing functional injection/extrusion), which may affect the material properties and some functional features, as reported by Tunesi et al. (NPG Asia Materials, 2019, 11(1), 28) independently from the kind of material and from hot or cold process. As the paper lacks in these features a brief introduction (e.g., 1-2 sentences) should be appreciated. Then, the authors should continue to stress their approach related to hot melt extrusion-triggered amorphization as a continuous process for inducing extended  supersaturable immediate-release of a BCS II drug  from supersaturating amorphous self-micellizing  solid dispersions. All of this should improve the quality of the paper, reporting important features as well as brief functional and technical considerations, thus helping the different kinds of readers to better understand the value of their work.
  • The Introduction section as well as the list of references should be improved according to the above reported comments.
  • The quality of some figures should be improved.
  • The title is adequate and appropriate for the content of the article.
  • The abstract contains information of the article.
  • Figures and captions are essential and clearly reported.

Reviewer 3 Report

Dear authors

You have developed a very interesting work with a lot of results a good explanations about them. Neverthless, I have some suggestions to do:

  • The self-micelling system is not explained in the sense to try to explain what you mentioned as "spring and parachute" technique. I thing that the use of this surfactant an ethoxilated alcohol, gives to help the existence of amorphous structures due to this micellization process. Please, consider to explain a little bit the relationship between these two phenomena involved. Specially what deals with HME system in which there are several steps that change the phycico-chemical behaviour of the mixture prepared before
  • Fig 1, in that sense would be interesting to be divided into the spring parachute description and the rest. It is too crowdy in informations. Please, consider to do it
  • In Line 112, you mentioned the use of Noyes-Whitney equations but these equations have been not used or described somewhere. If you will not use them, it is not necessary to be mentioned
  • Line 127. You mention PXRD-based stability whithout describing the power X-ray technique meaning anywhere
  • Line 278-283, and 291 to 302. This part of results is a reiteration of Introduction. Some terms are repeated and indeed some ideas. Please brig that part of text to Introudction paragraph and combine with the rest of content there, or eliminate from Results paragraph there is no need to repeat
  • Fig 3. There is an interesting change in the slope of samples until 150ºC. that is totally indicator of the presence of soluplus in them. Please, consider to add some comments about
  • Supersaturation is related with the capability of micelles formed to allocate the NIT molecules inside. When you discuss results from Fig 5, please, consider to add this aspect (results from Fig 11, suggest some interactions formed in the micellization process). This, can help also to explain what you named as inhibition of crystallization that is the stabilization of NIT molecules in more stable structures than to form crystals, although we are in the supersaturation level (Fig 12). Please, consider to relate the results shown in these figures.
  • When in lines 566-576 . How do you calculate the free energy of the system of the transition, and which is the transition you refer to?? Please, consider to add the equations used to calculate, as well as a description of the methodology used to calculate them. Results in this aspect are very important because they give, directly the stability of the structures.

Many thanks
